# Antimonotonicity, Hysteresis and Coexisting Attractors in a Shinriki Circuit with a Physical Memristor as a Nonlinear Resistor

Lazaros Laskaridis *, Christos Volos and Ioannis Stouboulos

Laboratory of Nonlinear Systems, Circuits & Complexity (LaNSCom), Department of Physics, Aristotle University of Thessaloniki, 54124 Thessaloniki, Greece; volos@physics.auth.gr (C.V.); stouboulos@physics.auth.gr (I.S.)
* Correspondence: llaskari@physics.auth.gr

**Abstract:** A novel approach to the physical memristor's behavior of the KNOWM is presented in this work. The KNOWM's memristor's intrinsic feature encourages its use as a nonlinear resistor in chaotic circuits. Furthermore, this memristor has been shown to act like a static nonlinear resistor under certain situations. Consequently, for the first time, the KNOWM memristor is used as a static nonlinear resistor in the well-known chaotic Shinriki oscillator. In order to examine the circuit's dynamical behavior, a host of nonlinear simulation tools, such as phase portraits, bifurcation and continuation diagrams, as well as a maximal Lyapunov exponent diagram, are used. Interesting phenomena related to chaos theory are observed. More specifically, the entrance to chaotic behavior through the antimonotonicity phenomenon is observed. Furthermore, the hysteresis phenomenon, as well as the existence of coexisting attractors in regards to the initial conditions and the parameters of the system, are investigated. Moreover, the period-doubling route to chaos and crisis phenomena are observed too.

**Keywords:** memristor; chaos; antimonotonicity; hysteresis; coexisting attractors; nonlinear resistor

## 1. Introduction

Leon Chua in 1971 postulated the fourth fundamental electrical component, which he called the memristor [1]. A memristor is a nonlinear circuit element linking the charge and the magnetic flux [2]. In contrast to a linear resistor, the memristor has a dynamic relationship between current and voltage, including a memory of past voltages or currents [3].

Some years later, Chua and Kang generalized the concept to memristive systems [4]. These systems are unconventional in the sense that while they behave like resistive devices, they can be endowed with a rather exotic variety of dynamic characteristics. Experimentally, the ideal memristor is yet to be demonstrated [5,6]. However, in 2008, a team from the Hewlett-Packard (HP) labs fabricated the first electronic passive memristor [7]. What is more, after this first attempt, also other research teams started to produce various types of memristors and make them commercially available, such as KNOWM Inc. (Santa Fe, NM, USA).

The memristor has the potential to augment or enhance several areas of integrated circuit design and computing. Extensive works in the literature regarding applications of the memristor have been produced since the HP announcement but only few developments are particularly noteworthy. One highly pervasive area where memristors may be applied is that of non-volatile random access memory (NVRAM) [8]. The memristor seems to have significant potential in this area, as the device exhibits memory, but it does not require continuous power draw and consumes little physical area. Furthermore, the memristor can be used for digital memory applications, where one bit of information can be stored using a single memristor. This can be achieved forcing the memristor to its extreme resistance

values ($R_{ON}$ and $R_{OFF}$), which correspond to either a 1 or a 0 [8]. Additionally, the memristors can be used as associative memories [9]. These memories map an input pattern to an output one according to the similarities of the input pattern to the pattern stored in the memory. Moreover, the memristor can be possibly used for nano-scale low power memory and distributed state storage, as a further extension of NVRAM capabilities. While the memristor can be used at its extreme resistance values in order to provide digital memory, it can also be made to behave in an analog manner. One potential application of this behavior is that of a dynamically adjustable electric load [10]. What is more, some other applications of memristors are in programmable logic [11], signal processing [12], super-resolution imaging [13], physical neural networks [14], control systems [15], reconfigurable computing [16], brain–computer interfaces [17] and RFID (radio-frequency identification) [18]. Additionally, memristive devices are used for stateful logic implication, allowing a replacement for CMOS (complementary metal-oxide-semiconductor) based logic computation [19]. Several early works have been reported in this direction [20,21]. However, because there are no effective memristors commercially available, only emulators [22,23] as well as ReRAM (resistive random-access memory) memristive models [24–26] are used.

Furthermore, the nonlinear characteristic behavior of the memristor has given to the research community the idea that it could be exploited in implementing novel chaotic circuits and systems with complex dynamics. In this direction, over the last three years, a few implementations of chaotic circuits with physical memristors have been proposed [24,27,28].

In this work, a different approach regarding the use of the KNOWM physical memristor [29], whose material stack is based on a mobile metal ion conduction through a chalcogenide material that has undergone a metal-catalyzed chemical reaction that creates channels, which constrain the flow of metal ions, is followed. It is experimentally observed that for low frequencies, the KNOWM memristor behaves approximately as a static nonlinear resistor. This drawback of the KNOWM memristor could be a real interesting feature, due to the fact that it could be used as a nonlinear resistor in chaotic circuits. Therefore, by using the experimental data of the KNOWM memristor's nonlinear *i-v* characteristic curve, the mathematical description of the nonlinear resistor is calculated. Next, the memristor as the proposed nonlinear resistor, is used in the well-known Shinriki chaotic oscillator circuit. Finally, the numerical investigation of the circuit's dynamics is presented. This investigation is based on the simulation results, which are produced by using numerical tools, such as phase portraits, maximal Lyapunov exponents [30–33], bifurcation diagrams [34], and continuation diagrams [35].

The paper is organized as follows. In Section 2, the mathematical model of the KNOWM memristor's characteristic curve for low frequencies, as well as the proposed chaotic circuit, are introduced. In Section 3, the numerical investigation of the circuit's dynamical behavior is presented. Finally, the conclusions and some thoughts for future works are discussed in Section 4.

## 2. Mathematical Model of the Chaotic Circuit

In this section, the Shinriki's chaotic circuit [36] is presented with some changes. The KNOWM memristor replaces the nonlinear positive conductance and at the same time, it is used as a nonlinear resistor ($N_R$). The schematic of the circuit is presented in Figure 1. The circuit consists of five linear resistors, two capacitors, one inductor and one operational amplifier. Moreover, the operational amplifier operates as a negative conductance. Additionally, the inductor $L$ and the capacitor $C_2$ consist of a resonant circuit.

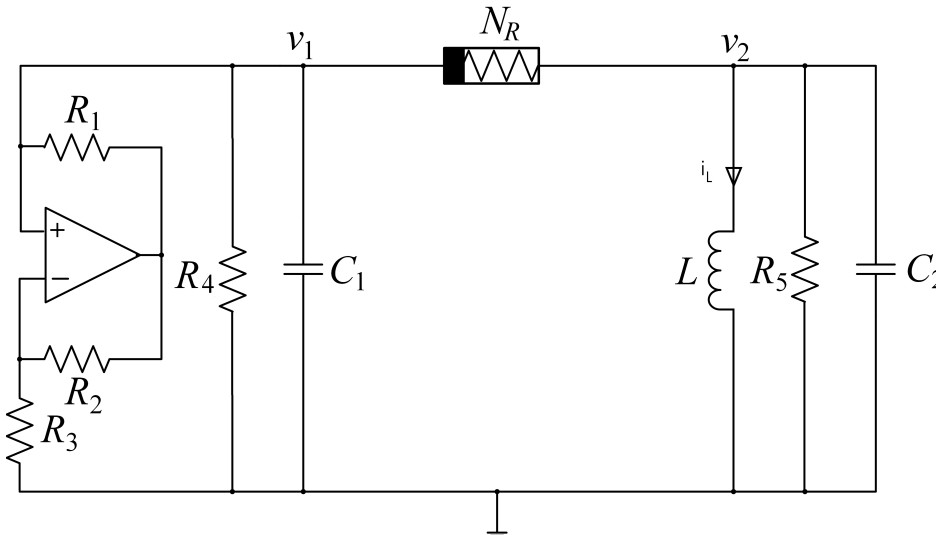

**Figure 1.** Shinriki's circuit with KNOWM memristor as a nonlinear resistor.

The KNOWM memristor used in the Shinriki circuit is one of the eight KNOWM memristors, which are contained in the 16-pin ceramic DIP (Dual inline package) package. In order to capture the characteristic $i_M - v_M$ of the selected memristor, the Analog Discovery 2 USB (Universal serial bus) oscilloscope is used by using the sinusoidal signal of amplitude 2.4 V and frequency 10 Hz. The curve is depicted in Figure 2. From this curve, it is obtained that the pinched hysteresis loop of the memristor's *i-v* characteristic curve shrinks so much that it could be considered a simple nonlinear curve.

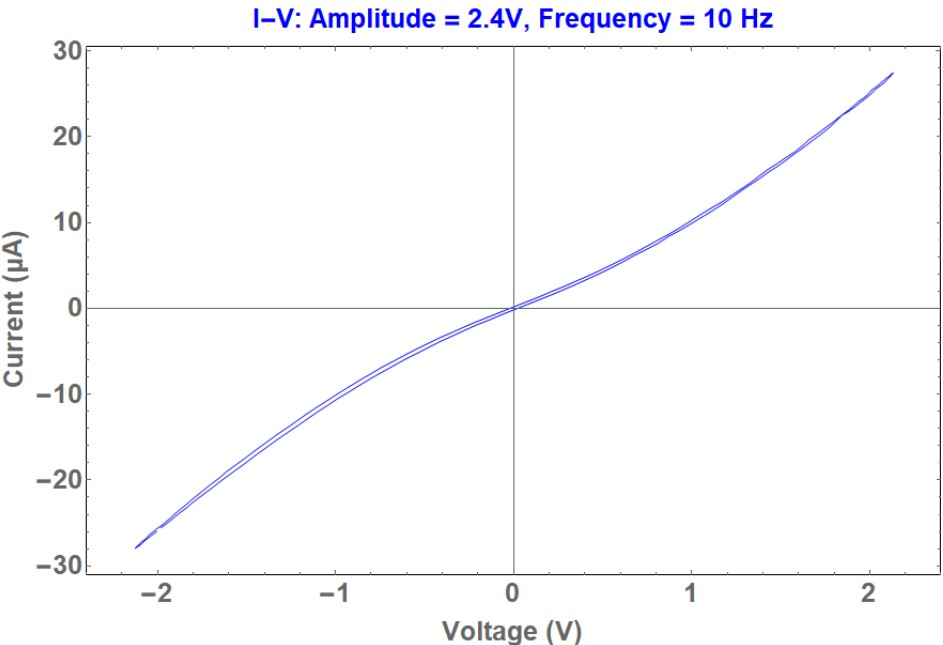

**Figure 2.** Experimental $i_M - v_M$ characteristic curve of the memristor.

Moreover, the mathematical formula of the memristor's characteristic $i_M - v_M$ curve is calculated by fitting the experimental data with the least squares method. Therefore, the following equation is produced:

$$i_M = 0.0125 \cdot e^{-0.00744 v_M} \cdot \sinh(0.68 \cdot v_M) \tag{1}$$

with $R^2 = 0.9996$, which is presented in Figure 3. So, for low frequencies, the KNOWM memristor could be used as a nonlinear resistor, for which its $i_M - v_M$ characteristic curve is described by Equation (1).

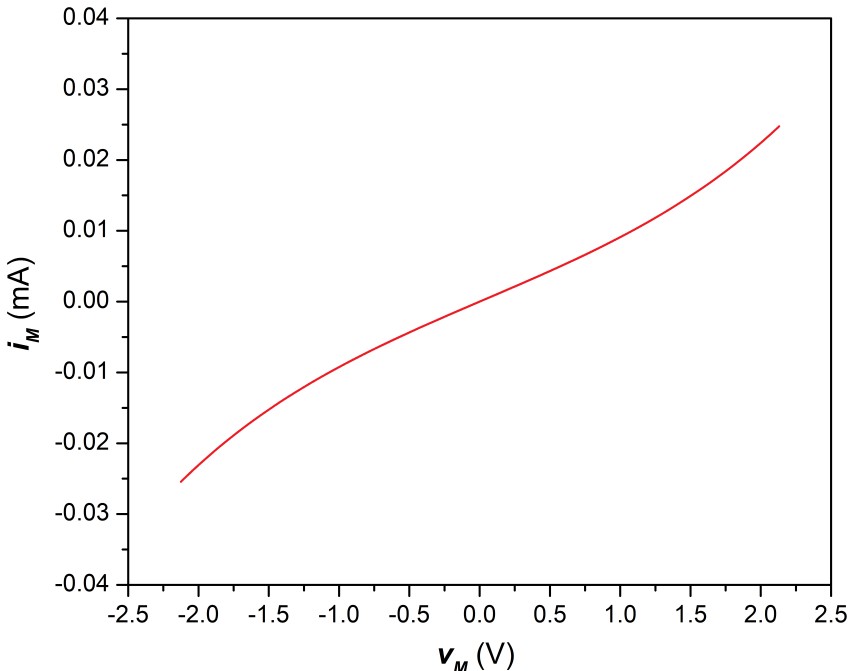

**Figure 3.** Fitting curve with Equation (1) of the experimental $i_M - v_M$ characteristic.

What is more, the resonant frequency of the circuit is given by

$$f_0 = \frac{1}{2\pi\sqrt{LC_2}} \tag{2}$$

In order to find the mathematical description of the proposed circuit, Kirchhoff's laws are applied to the circuit of Figure 1. Therefore, the dimensionless equations that describe the circuit are

$$
\begin{aligned}
\dot{x} &= \eta[(\alpha - \beta)x + i] \\
\dot{y} &= -z - \gamma y - i \\
\dot{z} &= y
\end{aligned}
\tag{3}
$$

where, $i$ represents the $i_M - v_M$ nonlinear characteristic curve of Equation (1). Additionally, the circuit's normalizing variables and parameters are

$$
x = \frac{v_1}{v_{ref}}, y = \frac{v_2}{v_{ref}}, z = \frac{\rho i_L}{v_{ref}}, i = \frac{\rho i_M}{v_{ref}}, \tau = \frac{t}{\sqrt{LC_2}}
$$

$$
\rho = \sqrt{\frac{L}{C_2}}, \eta = \frac{C_2}{C_1}, \alpha = \frac{\rho}{R_3}, \beta = \frac{\rho}{R_4}, \gamma = \frac{\rho}{R_5}
\tag{4}
$$

Next, in order to study the dynamical behavior of the circuit, the parameters of the system are set to the following values: $L = 0.5$ H, $C_2 = 506.66$ µF, $R_1 = R_2 = 5.6$ kΩ, and $R_3 = 0.109$ kΩ. The power supply is $\pm 10$ V.

## 3. Numerical Results

In this section, the dynamical behavior of the proposed, modified Shinriki circuit for different values of the linear resistance $R_5$ (parameter $\gamma$), the capacitance of the capacitor ($C_1$) and the initial conditions ($x_0, y_0, z_0$) is investigated. Generally, the system presents rich dynamics that include changes between regular and chaotic oscillations through the mechanism of crisis and period doubling phenomena.

### 3.1. Dynamics Related to the Capacitor $C_1$

In order to study the behavior of the system according to the capacitor $C_1$, bifurcation diagrams are produced. The value of the linear resistance is set to $R_5 = 0.1$ k$\Omega$. Figures 4–6 present the bifurcation diagrams of the variable $x$, with the capacitance $C_1$, for three different values of the linear resistance $R_4$ equal to 5, 10 and 15 k$\Omega$. Moreover, for the same values of the linear resistance $R_4$, the maximal Lyapunov exponent diagrams are presented in the same figures, respectively. From these diagrams, it is observed that there are regions where the system oscillates chaotically and regions where the system oscillates regularly. This behavior is also verified from the maximal Lyapunov exponent diagrams. In more detail, it is clear that when the Lyapunov exponent is positive, the existence of chaotic behavior is observed, while when the Lyapunov exponent is not positive, the system has a periodic behavior. Furthermore, as the value of the resistance $R_4$ increases, the chaotic behavior occurs in higher values of the capacitance $C_1$.

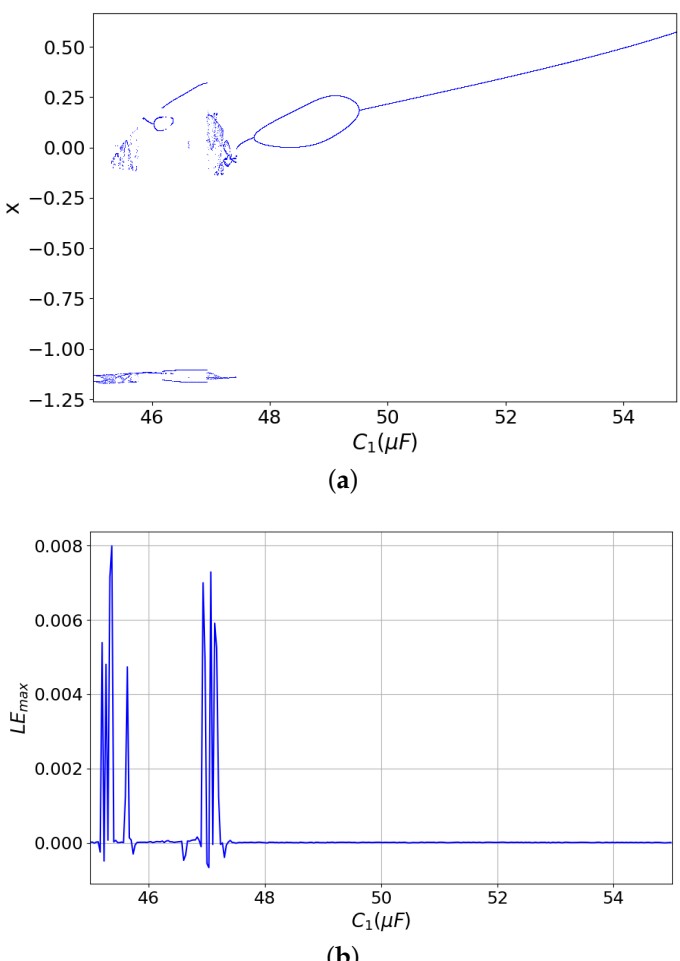

**(a)**

**(b)**

**Figure 4.** (**a**) Bifurcation diagram of $x$ versus $C_1$ and (**b**) maximal Lyapunov exponent diagram, for $R_4 = 5$ k$\Omega$.

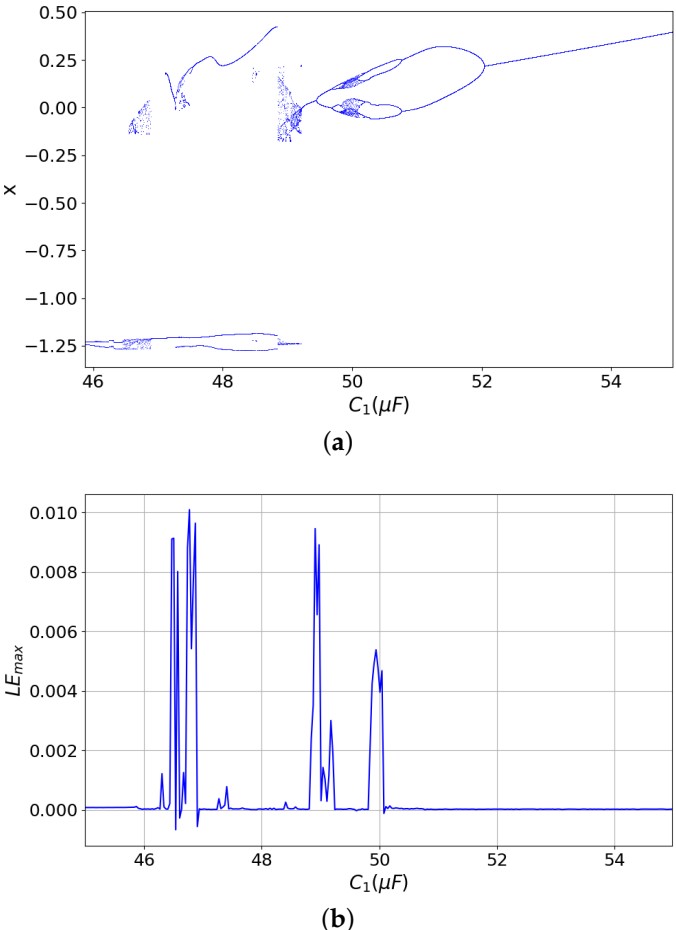

(**a**)

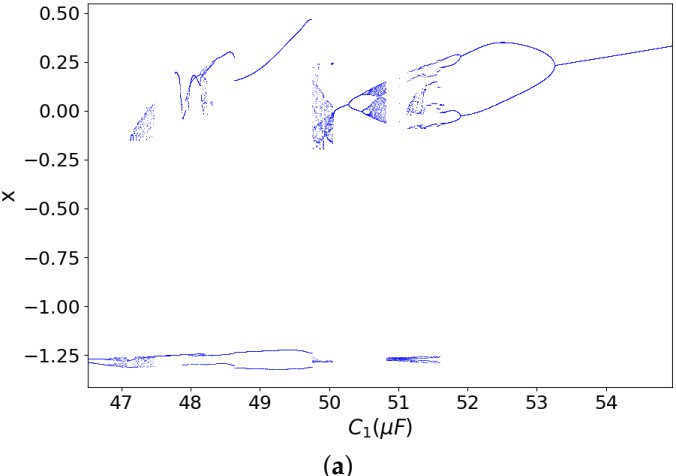

(**b**)

**Figure 5.** (**a**) Bifurcation diagram of $x$ versus $C_1$ and (**b**) maximal Lyapunov exponent diagram, for $R_4 = 10$ kΩ.

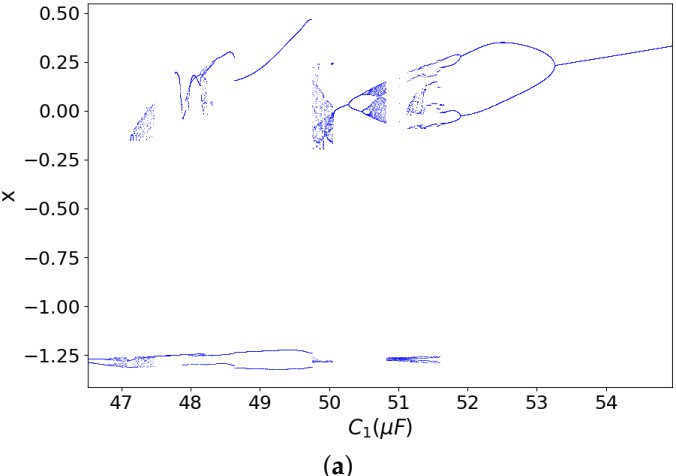

(**a**)

**Figure 6.** *Cont.*

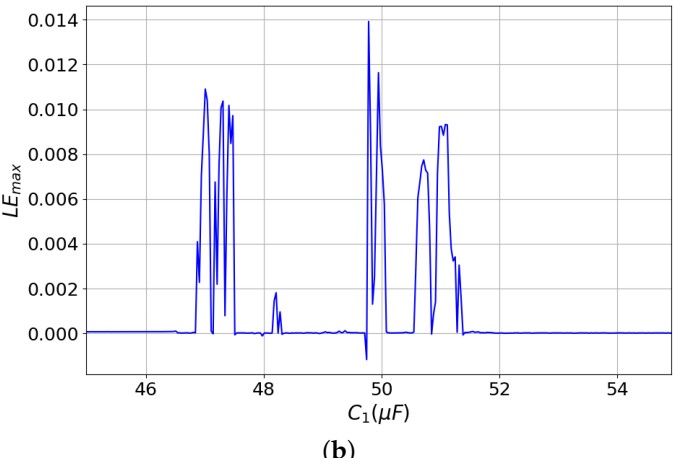

(**b**)

**Figure 6.** (**a**) Bifurcation diagram of $x$ versus $C_1$ and (**b**) maximal Lyapunov exponent diagram, for $R_4 = 20\,\mathrm{k\Omega}$.

Furthermore, from Figure 5, the antimonotonicity phenomenon [37–40] can be revealed. According to this phenomenon, the system enters into chaos with the well-known period doubling route and exits from the chaos following the reverse period doubling route. As a result, a shape of a chaotic bubble is formed in the bifurcation diagram. This phenomenon is relevant because it describes a complex scenario of how a nonlinear system creates or destroys unstable periodic orbits by parameter alterations. Thus, Figure 7 presents the bifurcation diagrams in regard to the parameter value $C_1$, as the linear resistance $R_4$ increases from $5\,\mathrm{k\Omega}$ to $11\,\mathrm{k\Omega}$. Moreover, in these diagrams, the bubble starts with period-1, but as the linear resistance $R_4$ increases, the period of the bubble increases too. Finally, a chaotic bubble is formed.

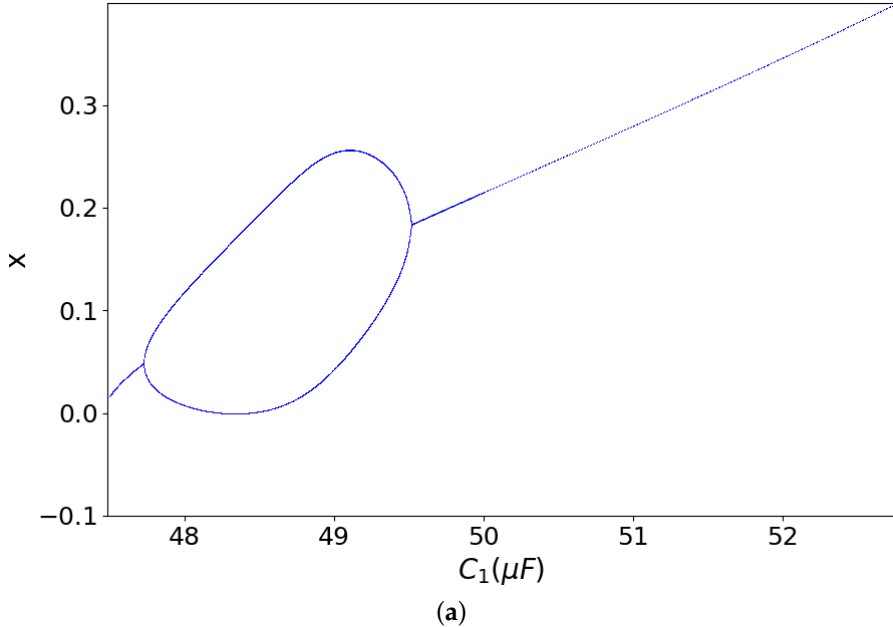

(**a**)

**Figure 7.** *Cont.*

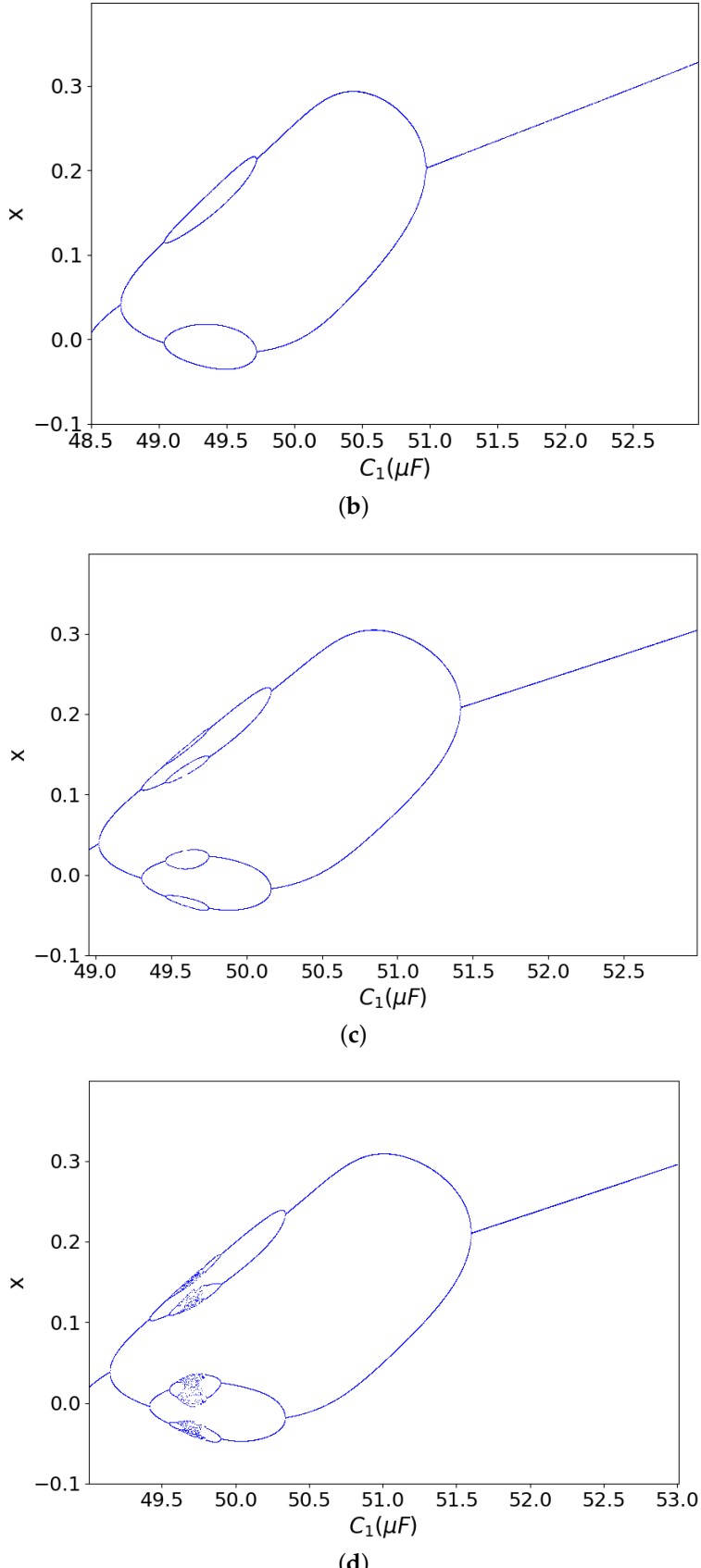

(**b**)

(**c**)

(**d**)

**Figure 7.** *Cont.*

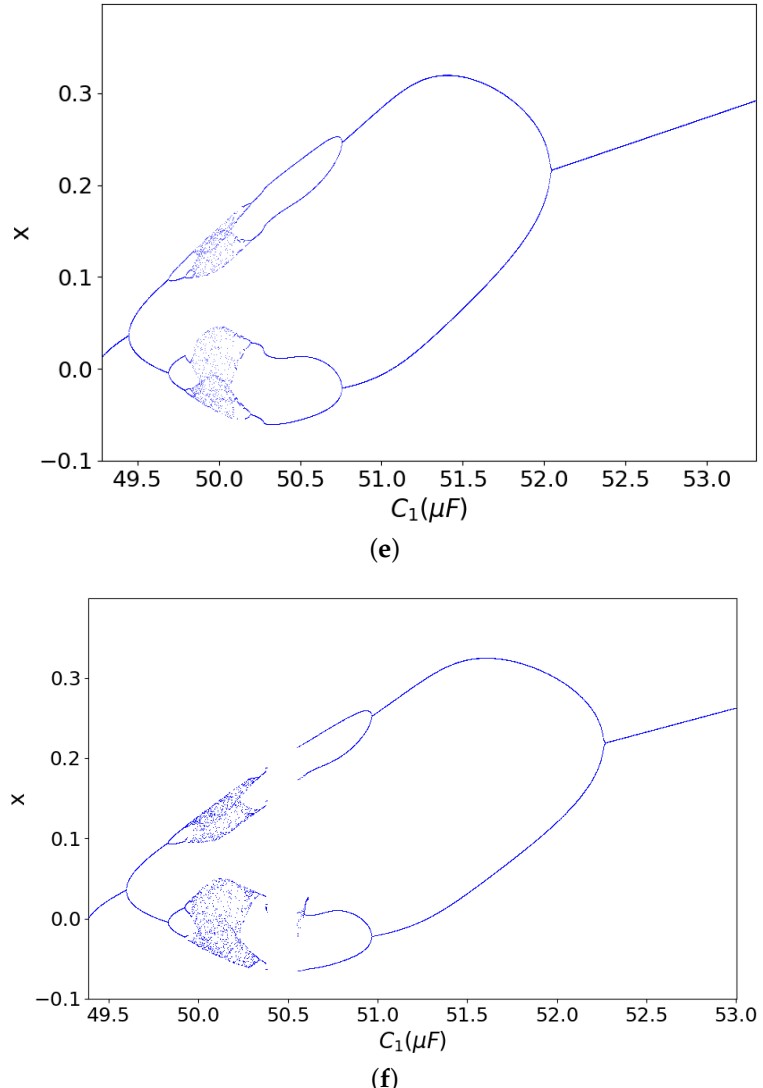

**Figure 7.** Bifurcation diagram of system, for: (**a**) $R_4 = 5$ kΩ, (**b**) $R_4 = 7$ kΩ, (**c**) $R_4 = 8$ kΩ, (**d**) $R_4 = 8.5$ kΩ, (**e**) $R_4 = 10$ kΩ and (**f**) $R_4 = 11$ kΩ.

### 3.2. Dynamics Related to the Initial Conditions

In this subsection, the dynamical behavior of the system is investigated in regard to the initial conditions $x_0, y_0, z_0$. More specifically, the parameters of the system are $R_4 = 20$ kΩ, $R_5 = 0.1$ kΩ and $C_1 = 50.66$ μF. So, in Figures 8–10, the bifurcation-like diagrams and the maximal Lyapunov exponent diagrams, in regard to the initial conditions $x_0, y_0$ and $z_0$, are produced. From these diagrams, it is observed that the dynamical behavior of the system changes in regard to the initial conditions $x_0, y_0$ and $z_0$.

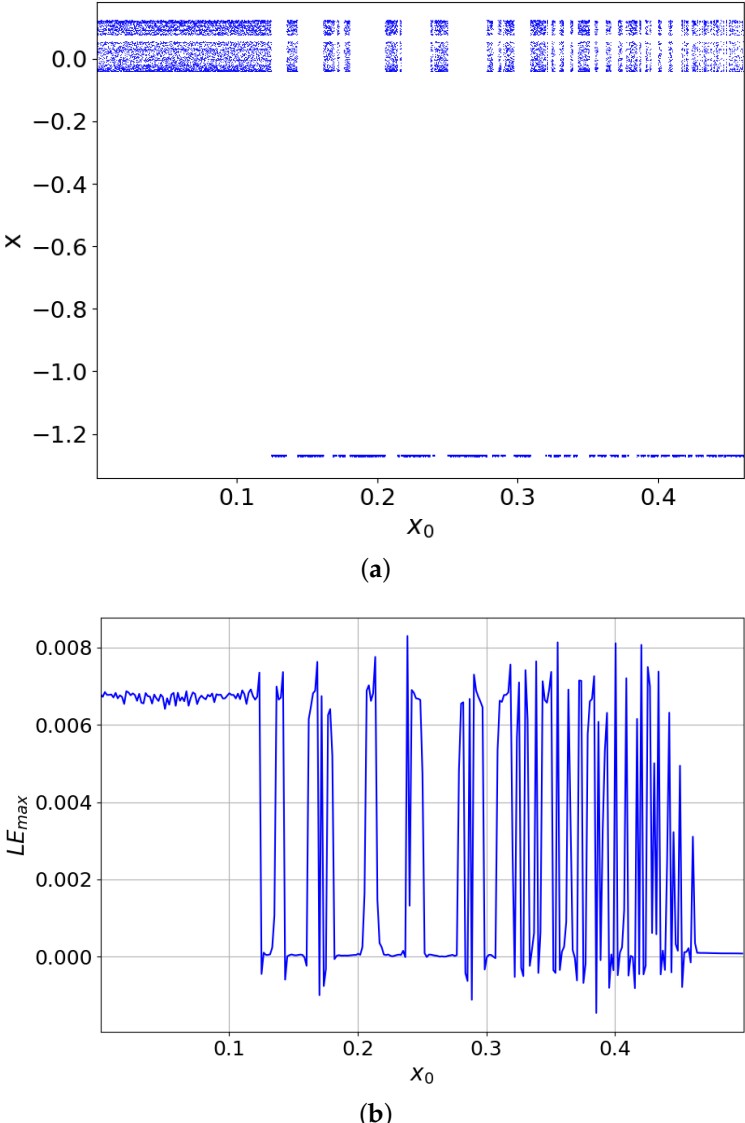

(a)

(b)

**Figure 8.** (**a**) Bifurcation-like diagram and (**b**) maximal Lyapunov diagram of system in regard to $x_0$, for $R_4 = 20$ kΩ, $R_5 = 0.1$ kΩ and $C_1 = 50.66$ μF.

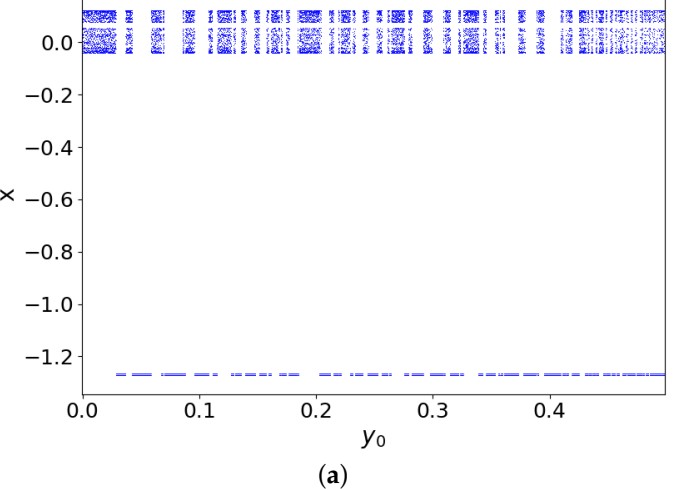

(a)

**Figure 9.** *Cont.*

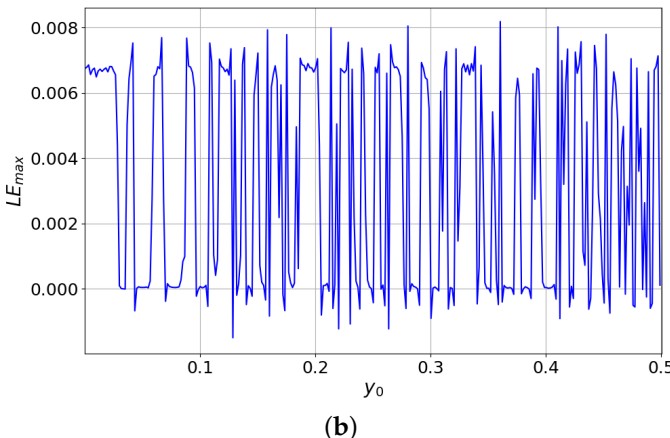

**(b)**

**Figure 9.** (**a**) Bifurcation-like diagram and (**b**) maximal Lyapunov diagram of system in regard to $y_0$, for $R_4 = 20$ kΩ, $R_5 = 0.1$ kΩ and $C_1 = 50.66$ μF.

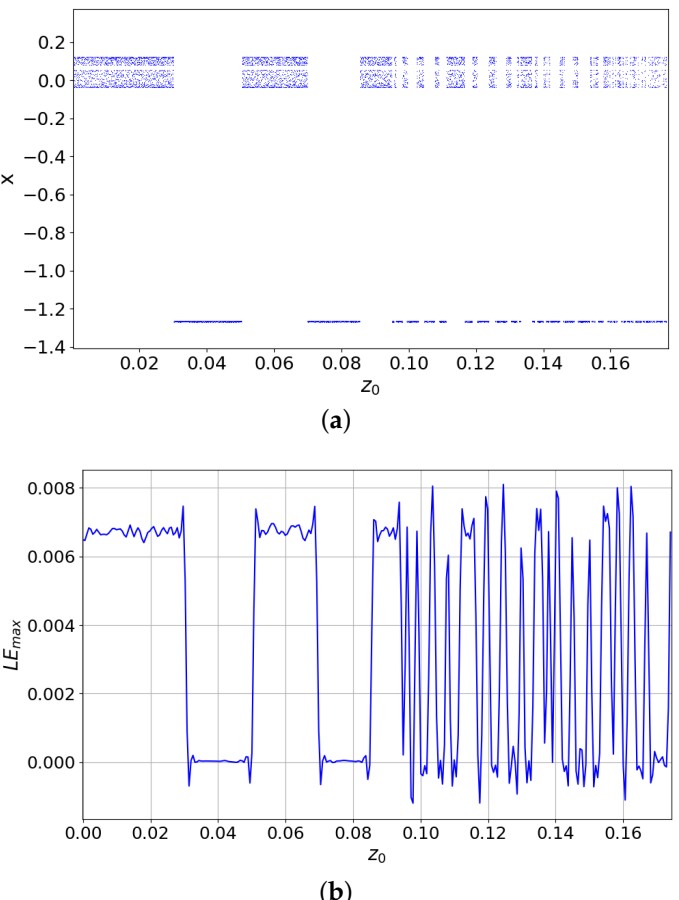

**(a)**

**(b)**

**Figure 10.** (**a**) Bifurcation-like diagram and (**b**) maximal Lyapunov diagram of system in regard to $z_0$, for $R_4 = 20$ kΩ, $R_5 = 0.1$ kΩ and $C_1 = 50.66$ μF.

More specifically, there are regions where the behavior is only chaotic ($0.0 < x_0 < 0.125$) and regions where the behavior is only regular ($0.22 < x_0 < 0.235$). Additionally, except for these two regions, the behavior of the system changes rapidly between chaotic and regular behavior, as it is observed from the bifurcation-like diagrams, as well as from the maximal Lyapunov exponents diagrams.

Next, in order to study more the effect of initial conditions in the dynamical behavior of the system, the basin of attraction of the system, in the $x_0 - y_0$ plane is produced, and it

is presented in Figure 11. From this figure, it is observed that for different initial conditions, the system behaves either regularly (blue points) or chaotically (red points). The basins of attraction diagram is constructed by separating the initial conditions to these, which have positive and no positive maximal Lyapunov exponents.

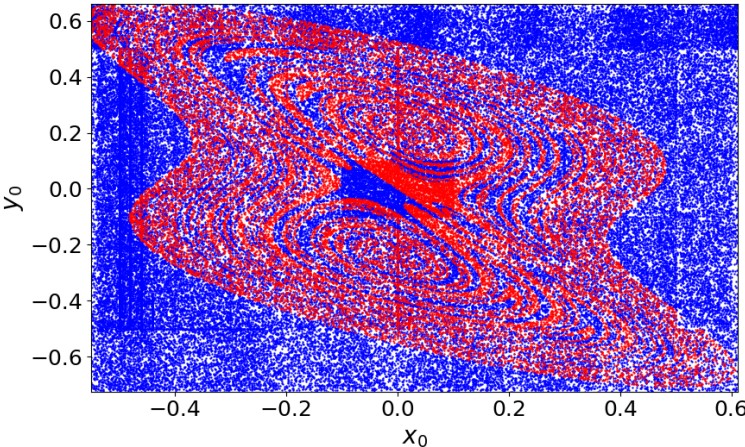

**Figure 11.** Basin of attraction of the system, for $R_4 = 20$ k$\Omega$, $R_5 = 0.1$ k$\Omega$ and $C_1 = 50.66$ μF.

### 3.3. Dynamics Related to the Parameter $\gamma$ (Linear Resistance $R_5$)

In this section, the numerical results from the simulations, regarding the value of the parameter $\gamma$, are presented. The bifurcation diagrams in regard to the parameter $\gamma$, for specific values of the linear resistance $R_4$ and capacitance of the capacitor $C_1$, are produced, and they are presented in Figures 12a–14a. From these diagrams, it can be easily observed that as the value of the linear resistance $R_4$ is low, the chaotic behavior dominates in lower values of the parameter $\gamma$, which means higher values of the linear resistance $R_5$. On the other hand, as the value of the parameter $R_4$ increases, the chaotic behavior occurs in all the range of the parameter $\gamma$. However, there are regions with regular behavior between chaotic oscillations. Moreover, in Figure 13a, for $\gamma = 0.2454$, a sudden jump from the upper part of the diagram to the lower part is observed. This phenomenon is known as hysteresis. Furthermore, these observations are also verified from the maximal Lyapunov exponent diagrams, which are presented in Figures 12b–14b.

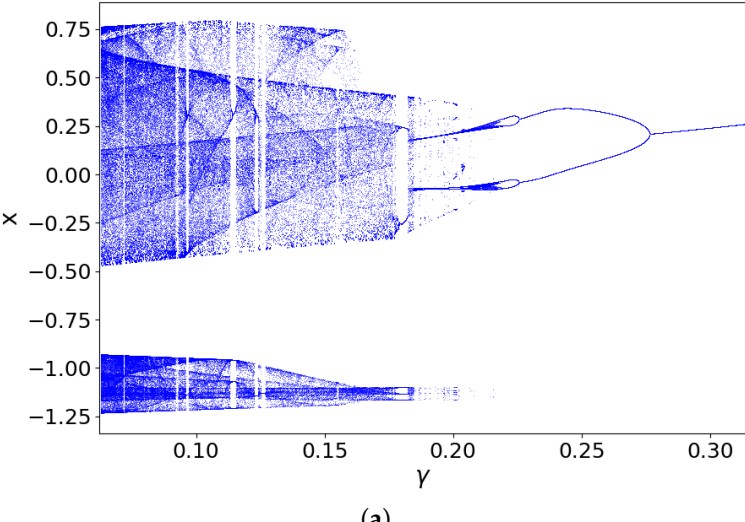

(**a**)

**Figure 12.** *Cont.*

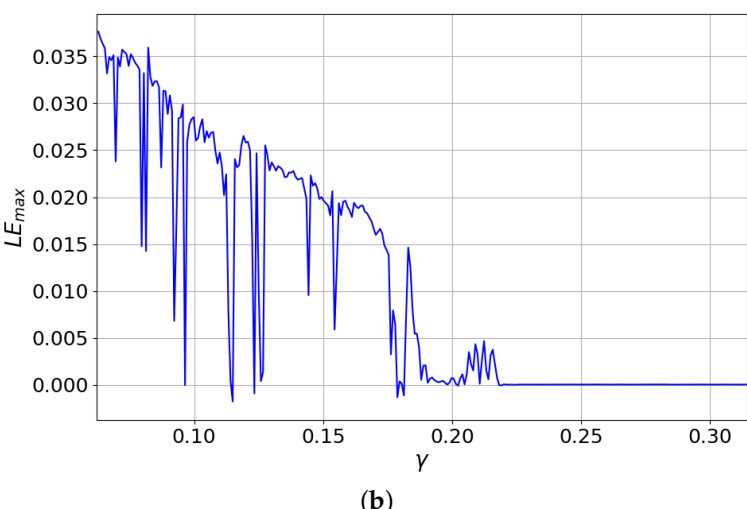

(**b**)

**Figure 12.** (**a**) Bifurcation-like diagram and (**b**) maximal Lyapunov diagram of system in regard to $\gamma$, for $R_4 = 5$ k$\Omega$.

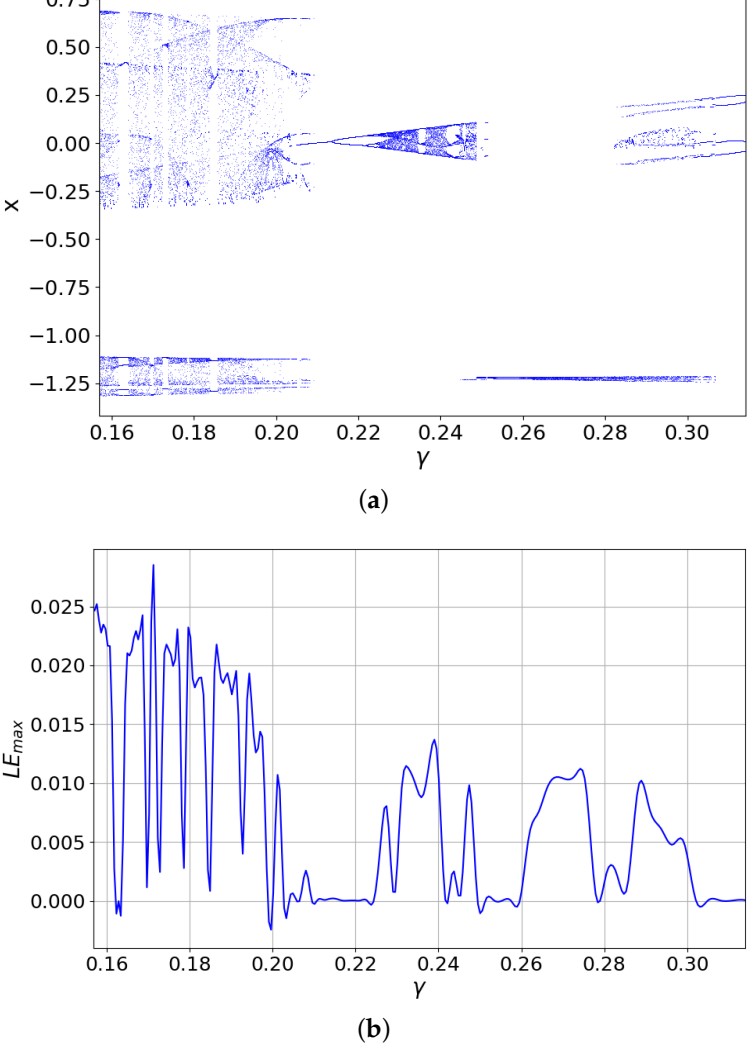

(**a**)

(**b**)

**Figure 13.** (**a**) Bifurcation-like diagram and (**b**) maximal Lyapunov diagram of system in regard to $\gamma$, for $R_4 = 10$ k$\Omega$.

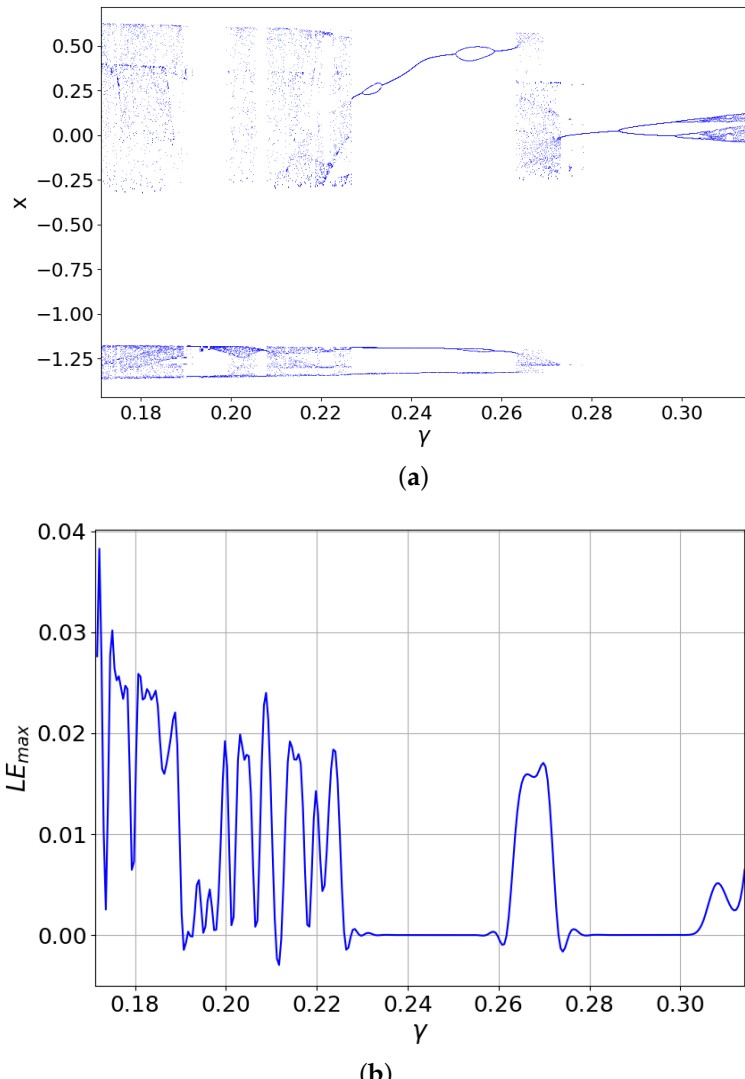

**Figure 14.** (**a**) Bifurcation-like diagram and (**b**) maximal Lyapunov diagram of system in regard to $\gamma$, for $R_4 = 20$ kΩ.

Next, the continuation diagrams with the parameter $\gamma$ are constructed for the values of the linear resistance $R_4 = 5$ kΩ and $R_4 = 10$ kΩ. The difference with the corresponding bifurcation diagrams is that in the continuation diagrams, the final values of the variables $x$, $y$ and $z$, for a specific value of the parameter $\gamma$ are used as the next initial conditions for the next value of the parameter $\gamma$. The continuation (with red color) and bifurcation (with blue color) diagrams, for $R_4 = 5$ kΩ and $R_4 = 10$ kΩ are presented in Figure 15. Comparing the bifurcation with the continuation diagram, the phenomenon of coexisting attractors can be revealed. Two coexisting attractors (chaotic and periodic), for $\gamma = 0.24$ and $R_4 = 10$ kΩ are presented in Figure 16 respectively.

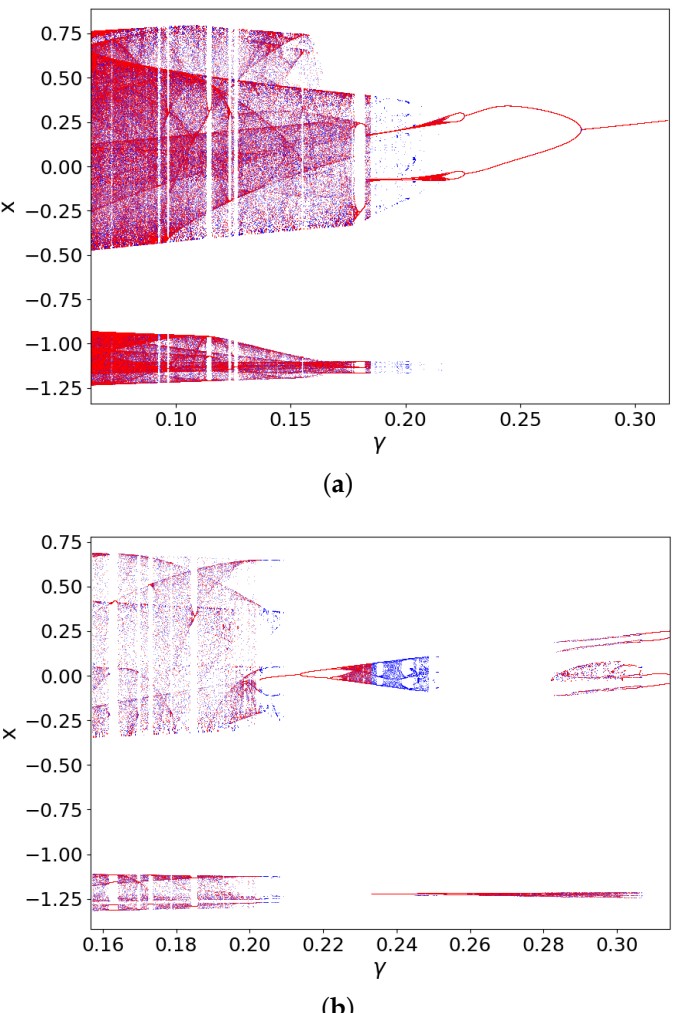

(**a**)

(**b**)

**Figure 15.** Continuation (red) and bifurcation (blue) diagram of parameter $\gamma$, for (**a**) $R_4 = 5\,\text{k}\Omega$ and (**b**) $R_4 = 10\,\text{k}\Omega$.

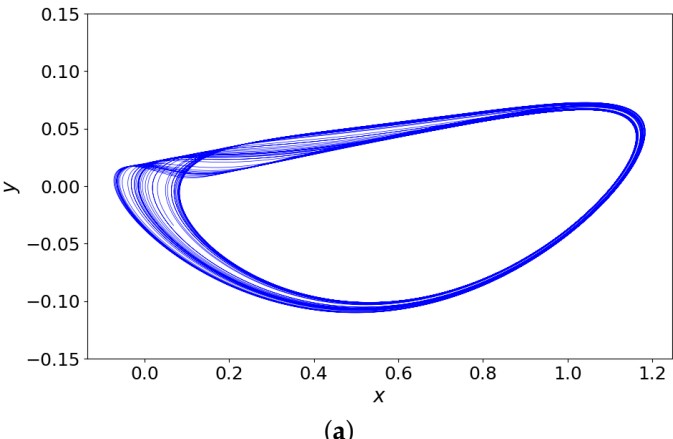

(**a**)

**Figure 16.** *Cont.*

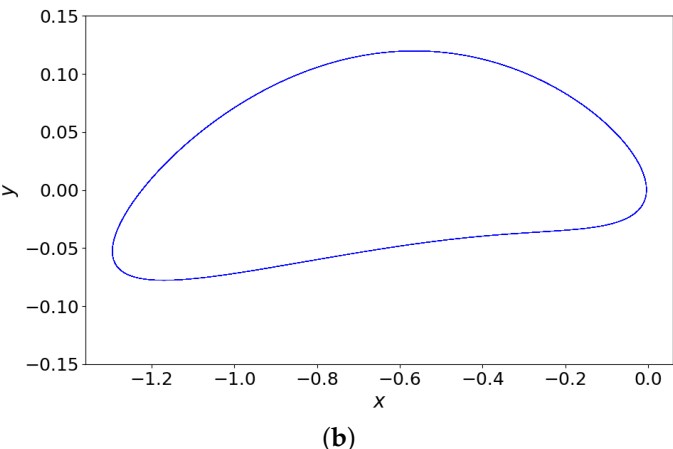

**(b)**

**Figure 16.** Phase space in $x$-$y$ plane, for $\gamma = 0.24$ and for (**a**): $x_0 = 0.05$, $y_0 = 0.01$, $z_0 = 0.0$ and (**b**) $x_0 = -1.119915$, $y_0 = -0.076945$, $z_0 = -0.276554$.

## 4. Conclusions

In this work, an autonomous chaotic circuit with a physical memristor was studied. The idea was to study the dynamical behavior of the circuit by using a KNOWM memristor as a nonlinear resistor. Plenty of numerical tools in order to study the circuit's dynamical behavior, such as the phase portraits, the bifurcation and continuation diagrams, as well as the diagrams of maximal Lyapunov exponent, were used.

The autonomous system (3) presented rich dynamical behavior. More specifically, chaotic and regular behavior were observed. Moreover, the system presented a route to chaos through the mechanism of period doubling, as well as crisis phenomena. From the bifurcation diagrams in regards to the capacitance $C_1$, for different values of the linear resistance $R_4$, it is observed that as the resistance $R_4$ increases, the chaotic behavior covers a larger range of the values of the capacitance $C_1$. Furthermore, with the increase in the resistance $R_4$, the antimonotonicity phenomenon occurs.

The second approach in this work was to study the system's behavior according to the change of initial conditions $x_0, y_0$ and $z_0$. In more detail, the system's behavior was chaotic with the existence of periodic windows inside chaotic regions. Moreover, from the basins of attraction diagram, a symmetric dynamical behavior in regards to the initial conditions was observed.

Furthermore, the third approach in this work was to change the parameter $\gamma$, which corresponds to the linear resistance $R_5$, and the system's behavior for different values of $R_4$ was studied. As well as, in the first approach, as the linear resistance $R_4$ increases, the chaotic behavior occurred in all the range of the parameter $\gamma$. Furthermore, the comparison of continuation and bifurcation diagrams revealed the existence of coexisting attractors, as well as the hysteresis phenomenon.

Therefore, this work presents the usefulness of the memristor as a nonlinear resistor in chaotic circuits. More specifically, some possible applications of the proposed circuit could be in chaotic encryption, as well as in the design of secure chaotic communication systems. Finally, a further study of this work will be the experimental implementation of the proposed circuit in order to verify its dynamical behavior.

**Author Contributions:** Conceptualization, L.L., C.V. and I.S.; methodology, L.L., C.V. and I.S.; software, L.L.; validation, L.L., C.V. and I.S.; formal analysis, L.L., C.V. and I.S.; writing—original draft preparation, L.L.; writing—review and editing, L.L., C.V. and I.S. All authors have read and agreed to the published version of the manuscript.

**Funding:** This research received no external funding.

**Conflicts of Interest:** The authors declare no conflict of interest.

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
