# Peer review of "Antimonotonicity, Hysteresis and Coexisting Attractors in a Shinriki Circuit with a Physical Memristor as a Nonlinear Resistor"

_electronics, doi:10.3390/electronics11121920_

Round 1

Reviewer 1 Report

1. In the introduction, 1-3 paragraphs explain the history of memristor but it should be simplified and merged. 

2. Generally, texts in graphs are low visibility so those should be improved.

3. The format of references should be unified.
ex ) 2. Tetzlaff, R. Memristors and memristive systems; Springer, 2013. 

Author Response

Thank you for your comments.

Reviewer 2 Report

The authors Lazaros Laskaridis, Christos Volos,  and Ioannis Stouboulos have submitted a manuscript entitled " Antimonotonicity, Hysteresis and Coexisting Attractors in a Shinriki Circuit with a Physical Memristor as a Nonlinear Resistor " to the journal Electronics.

The introduction provides sufficient background and includes all relevant references. The research design is appropriate. The methods are adequately described. The results are clearly presented. Discussion of data and conclusions are adequately supported by the results.

English language and style are minor spell check required.

I do not detect plagiarism and I do not detect inappropriate citations.

In general, I do not see any ethical issues along the manuscript

The manuscript is interesting.

I have few comments:

1) I would suggest to avoid acronyms in the abstract. Moreover, many acronyms are not explained: KNOWM, NVRAM, etc.

2) I think it is better to explain again the acronym in the caption of Figure 1.

3) Figure 2 seems to be a screen shot of the software of the experimental setup. The readability of the figure is very low. I suggest the authors to make a new figure with large font size etc.

4) Many figures have small font size in the labels. I suggest to increase the size.

5) Figures 9 - 16 are placed between the references. Thi will not help the reader.

6) I suggest to rephrase the sentence "Finally, as a further study of this work will be examined the implementation of the proposed circuit and the experimental confirmation of the numerical results." in the conclusion. The sentence is a bit cryptic.

Moreover, I would add a paragraph on possible applications.

Author Response

Thank you for your comments.

Round 2

Reviewer 2 Report

The authors have provided a revised version of the manuscript "Antimonotonicity, Hysteresis and Coexisting Attractors in a Shinriki Circuit with a Physical Memristor as a Nonlinear Resistor" for the journal Ellectronics. Together with the revised manuscript they have provided a point to point response letter in which they have addressed the comments.

In my opinion, the manuscript is improved with respect to the original submission. The authors have performed several revisions and integrations. 

The introduction provides sufficient background and includes all relevant references. The research design is appropriate. The methods are adequately described. The results are clearly presented. Discussion of data and conclusions are adequately supported by the results.

English language and style are minor spell check required.

I do not detect plagiarism and I do not detect inappropriate citations.

In general, I do not see any ethical issues along the manuscript.

In terms of originality, significance of content, quality of presentation, scientific soundness, interest to the readers, I think that the manuscript deserves publication in the journal Electronics. For this reason, I recommend the editorial board to accept the manuscript in the present form.